# To Balloon or Not to Balloon? The Effects of an Intra-Aortic Balloon-Pump on Coronary Artery Flow during Extracorporeal Circulation Simulating Normal and Low Cardiac Output Syndromes

**DOI:** 10.3390/jcm10225333

**Published:** 2021-11-16

**Authors:** Philippe Reymond, Karim Bendjelid, Raphaël Giraud, Gérald Richard, Nicolas Murith, Mustafa Cikirikcioglu, Christoph Huber

**Affiliations:** 1Charles Hahn Hemodynamic Propulsion Laboratory, Medical Faculty, University of Geneva, 1211 Geneva, Switzerland; Gerald.Richard@hcuge.ch (G.R.); Nicolas.Murith@hcuge.ch (N.M.); Mustafa.Cikirikcioglu@hcuge.ch (M.C.); 2Division of Cardiovascular Surgery, Department of Surgery, University Hospitals of Geneva, 1211 Geneva, Switzerland; 3Department of Anesthesiology, Pharmacology and Intensive Care, Geneva Hemodynamic Research Group, University Hospitals and Medical Faculty of Geneva, 1211 Geneva, Switzerland; Karim.Bendjelid@hcuge.ch (K.B.); Raphael.Giraud@hcuge.ch (R.G.)

**Keywords:** ECMO, IABP, in vitro mock-up circuit, low cardiac output syndrome, coronary blood flow

## Abstract

ECMO is the most frequently used mechanical support for patients suffering from low cardiac output syndrome. Combining IABP with ECMO is believed to increase coronary artery blood flow, decrease high afterload, and restore systemic pulsatile flow conditions. This study evaluates that combined effect on coronary artery flow during various load conditions using an in vitro circuit. In doing so, different clinical scenarios were simulated, such as normal cardiac output and moderate-to-severe heart failure. In the heart failure scenarios, we used peripheral ECMO support to compensate for the lowered cardiac output value and reach a default normal value. The increase in coronary blood flow using the combined IABP-ECMO setup was more noticeable in low heart rate conditions. At baseline, intermediate and severe LV failure levels, adding IABP increased coronary mean flow by 16%, 7.5%, and 3.4% (HR 60 bpm) and by 6%, 4.5%, and 2.5% (HR 100 bpm) respectively. Based on our in vitro study results, combining ECMO and IABP in a heart failure setup further improves coronary blood flow. This effect was more pronounced at a lower heart rate and decreased with heart failure, which might positively impact recovery from cardiac failure.

## 1. Introduction

Extracorporeal membrane oxygenation (ECMO) is the most frequently used mechanical circulatory support system in patients with low cardiac output syndrome who are refractory to medical treatment. The additional use of an Intra-Aortic Balloon Pump (IABP) in ECMO-supported patients might have hemodynamic and coronary blood flow (CBF) benefits by restoring systemic pulsatile flow conditions [1]. Although there is theoretical evidence in favor of adding IABP support to ECMO for improving coronary artery blood flow, the review studies report contradictory results [2,3,4,5,6]. Nevertheless, recent data suggest that the IABP has a clear beneficial effect on hemodynamic parameters in the non-acute coronary syndrome cardiogenic shock group [7].

This lack of scientific consensus prompted us to evaluate the effect of combining IABP and ECMO on coronary artery blood flow in a high-fidelity, in vitro, mock-circuit setup.

## 2. Materials and Methods

### 2.1. Systemic Circulation Model

A silicon phantom model reproducing the systemic circulation from the aortic root level to the iliac arteries was used (Figure 1). This silicon phantom replicates the vessel wall mechanical compliance of a healthy subject (Model ref T-S-N-009+, Elastrat, Geneva, Switzerland).

We added a compliance reservoir, made with a cylindrical reservoir of polymethyl methacrylate (PMMA) upstream to the silicon phantom to mimic the compliance (Cres) of the remaining arteries, which were not included in our silicon model. The volume of air (V_0_) and the absolute pressure (P_0_) on top of the glycerol/water mixture (Figure 2) in the cylindrical reservoir were adjusted to mimic the total compliance of the systemic circulation [8].

To replicate physiological cardiac output (CO), the in vitro model was connected to a piston pulsatile pump (Superpump, ViVitro Labs, Inc., Victoria, BC, Canada) that generated a flow waveform similar to that of a healthy subject. This generic flow waveform is based on averaged flow measurements in the ascending aorta of volunteers using in vivo phase contrast MRI (PC-MRI) flow quantification [8].

The distal ends of the silicon phantom perfusing the cerebral circulation were connected to an adjustable resistance to maintain an average flow of 700–800 mL/min, corresponding to the average values reported in the literature (mean 12.5 mL/s, range 10.3–12.6 mL/s) [8]. The distal ends of the model were adjusted with a fixed resistance (Hoffmann clamp) to reproduce a mean central arterial pressure.

All the outlets of the circuit emptied into an open reservoir to mimic the venous side of the circulation. The circuit was filled with a 40% glycerol/60% water mixture to mimic the blood’s rheological properties in terms of density and dynamic viscosity, estimated at 1106 (kg/m^3^) and 0.003 (Pa·s), respectively, according to Cheng [9].

Different clinical scenarios, such as normal cardiac output and moderate-to-severe heart failure (HF), were simulated by decreasing the pulsatile pump output flow (CO) from 5-3-2 L/min.

### 2.2. Coronary Artery Resistance Model

A coronary artery model with systolo-diastolic resistance synchronized to pump pulsatility was built to simulate the effect of left ventricular contraction. LV contraction generates varying transmural pressure on the coronary arteries, therefore varying resistance. We were able to simulate this varying resistance with a flexible silicon tube of low radial stiffness, where the surrounding pressure was the pulsatile pump chamber corresponding to the LV. The features of the waveform we obtained were similar to those published in the literature [10].

### 2.3. Extracorporeal (Membrane Oxygenation)

We utilized a veno-arterial extracorporeal membrane oxygenation (VA-ECMO) Rota-Flow system (Maquet GetingeGroup, Wayne, NJ, USA) without the membrane oxygenation component. The venous cannula drained the open venous reservoir and the centrifugal pump injected directly into the femoral artery of the model via a silicon introducer and a 19 Fr. cannula. In heart failure scenarios, peripheral ECMO support (+2 and +3 L/min) is used in order to compensate for low cardiac output values in order to achieve a default value of 5 L/min.

### 2.4. Intra-Aortic Balloon Counter-Pulsation

An Intra-Aortic Balloon Pump (IABP) catheter was placed at the thoracic aorta level (MEGA 8 Fr-50 cc, Datascope Corp., Maquet GetingeGroup) via an introducer at the iliac artery level. The IABP was connected to the Maquet CS100 Console set to semi-automatic mode 1:1, 100% volume amplitude, and triggered by the pressure wave (pressure sensor in the ascending aorta). Because HR affects the hemodynamic response of IABP, the effect of IABP on coronary blood flow was assessed during each of the clinical scenarios at an HR of 60 and then 100 bpm [11].

### 2.5. Pressure and Flow Measurements

The pressure waveform measurements were obtained with a Millar Mikro-Tip catheter MPC-500 (Millar Instruments, Houston, TX, USA) placed in the aortic arch.

Flow waveform measurements at the aortic valve and left coronary artery level were performed with an ME19PXN and an ME4PXN Transonic flow sensor (Transonic System Inc., Ithaca, NY, USA) connected to a Transonic T402-TB meter. The flow sensors were calibrated by the company for the given glycerol/water mixture ratio and temperature.

The waveforms were recorded during 8–10 consecutive cardiac cycles. The trigger at the beginning of each pump signal was used to synchronize each cardiac cycle waveform, allowing the calculation of the average curve along one cardiac cycle and facilitating further statistical analysis.

### 2.6. Data Acquisition and Post-Processing

Data acquisition of the pressure and flow waveform was performed with a PowerLab 8/35 acquisition card (ADInstruments, Oxford, UK) at a sampling frequency of 1 kHz. The post-processing of the pressure and flow waveforms was performed with Matlab R2019a (Mathworks, Natick, MA, USA). The waveforms were averaged over 8–10 consecutive cardiac cycles.

## 3. Results

Pressure waveforms using the Millar catheter at the ascending aorta level are presented for a baseline condition (CO 5 L/min) (Figure 3A), and also for a severe HF scenario (CO 2 L/min) (Figure 3C). The effects of the IABP on the pressure waveforms can be readily seen for the baseline condition (Figure 3B) and for the severe HF scenario (Figure 3D). Table 1 and Table 2. IABP increased the pressure during the diastolic phase while decreasing it in the early to middle part of the systolic phase of the cardiac cycle.

Measurements of coronary artery flow waveforms are presented for the baseline condition (CO 5 L/min) (Figure 4A) and the severe HF scenario (CO 2 L/min) (Figure 4C). The effects of the IABP on coronary artery flow waveforms are shown for the baseline condition (Figure 4B) and the severe HF scenario (Figure 4D). Table 1 and Table 2.

Figure 5A shows coronary mean flow in the main branch of coronary arteries at the baseline condition (CO 5 L/min), intermediate (CO 3 L/min), and severe (CO 2 L/min) levels of LV failure at 72.0 (+16%), 66.1 (+7.5%), and 60.0 (+3.4%) mL/min (% of difference when using IABP compared to condition without IABP) for an HR of 60 bpm. Coronary flow rates for the same scenarios but for an HR of 100 bpm are 77.4 (+6%), 69.0 (+4.5%), and 69.1 (+2.5%) mL/min are shown in Figure 5B. The increase in coronary blood flow using the combined IABP-ECMO setup was more noticeable in low HR conditions.

Enabling the IABP in the baseline condition (CO of 5 L/min, HR 60 bpm) and in the severe HF scenario (CO 2 L/min, HR 60 bpm) modified the pressure waveforms in the ascending aorta and is presented in Figure 6. The superposed pressure waveforms when IABP was enabled highlight the decrease in pressure during systole and an increase during the diastolic phase of the cardiac cycle.

The difference in the area under the curves (AUC) of pressure waveforms during the diastolic relative to the systolic phase illustrates the flow component contribution of IABP use.

## 4. Discussion

Based on our in vitro study results, reduced coronary artery blood flow in an experimental heart failure scenario can be compensated by ECMO support, explaining why ECMO support remains the mainstay treatment in acute cardiogenic shock. Our study shows the additional benefits of combining IABP with ECMO in further improving coronary artery blood flow.

ECMO is an external cardiovascular mechanical pump that drains blood from a venous cannula (venous side), where it is passed through a membrane for oxygenation and CO_2_ removal and then is subsequently re-injected into the arterial side of the systemic circulation through an arterial cannula.

Deflation of the IABP in the systolic phase decreases pressure in the aorta, thus enhancing the emptying of the left ventricle (LV). The quick inflation in the diastolic phase increases blood flow in the coronary arteries. The combined effect of IABP leads to an increase in stroke volume (SV), cardiac output (CO), and coronary artery blood flow (CBF) [1].

IABP is occasionally used in the clinical setting for patients already undergoing treatment with venous–arterial extracorporeal membrane oxygenation (VA-ECMO) exhibiting refractory cardiogenic shock (severe myocardial contractile dysfunction and tissue hypoperfusion) [1].

However, despite the theoretical benefit related to combined IABP/ECMO therapy in refractory cardiogenic shock, there is no consensus on its use in the clinical setting. The main results of several recent clinical studies are summarized below.

A recent systematic meta-analysis review [3], including 16 studies and 1517 patients, reported a cumulative survival rate of 37.5% when ECMO was utilized alone, compared to a 35.3% cumulative survival rate with IABP/ECMO combined. They concluded that survival was not improved with combined use and also not improved in cases when IABP therapy was started before ECMO, such as in cases of acute myocardial infarction (AMI). Other authors [5] reported on an observational cohort study of 529 patients (ECMO and ECMO + IABP). The primary outcome of mortality rates at 2 weeks was identical in both these groups, and there was no improvement (decreased incidence of multi organ failure) in patients with cardiopulmonary failure.

Further, a meta-analysis combining 22 studies and 4653 patients [6] considering short-term mortality for patients with cardiogenic shock undergoing VA-ECMO with or without the addition of IABP, concluded that there were no significant differences with combined vs. standard ECMO use. However, they highlighted a lower mortality rate in an AMI subgroup (50.8 vs. 62.4%) when IABP was added to ECMO. The present findings could be related to the fact that, by decreasing LV end-diastolic pressure following an unloading of the LV, IABP both decreases LV wall tension-LV transmural pressure and concomitantly increases coronary perfusion [12].

From a pathophysiological perspective, IABP has proven effective in low cardiac output syndromes. To date, the IABP has been clinically used due to its benefits in concomitantly decreasing LV afterload and improving coronary artery and systemic tissue perfusion [12]. This, in turn, results in a reduced need for inotropic and vasopressor drug therapy [13].

In this regard, a recent systematic meta-analysis reviewing 29 studies, including 4576 patients [4], compared a group of patients with IABP and ECMO to a group with ECMO alone. They reported that the addition of IABP could be associated with lower in-hospital mortality (primary outcome) without increasing neurological, gastrointestinal, or limb complications (secondary outcome).

To date, there appears to be no clear answer on the clinical benefits of the ECMO-IABP combination for patients, as well as no documented study quantifying the influence of this combination on coronary artery flow. For this reason, we were prompted to conduct this study to evaluate the theoretical hemodynamic benefits of combined IABP-ECMO use on coronary artery blood flow in an in vitro, mock-circuit setup.

This allowed us to deduce the following main points:(1)The performance of an IABP is dependent on heart rate in the absence or presence of ECMO [11]. The increase in coronary artery flow was more important in low heart rate conditions (HR 60 bpm). This would seem rational, as the time of balloon inflation is longer at a lower heart rate (time constant of duration) [14]. Indeed, the diastolic times in our model scenarios were about 650 and 390 ms for heart rates of 60 bpm and 100 bpm, respectively. Additionally, the dynamic effects during balloon inflation/deflation probably play a role at a higher HR, as coupling and resistance to retrograde flow from the femoral arteries (ECMO re-injection flow) may be increased.(2)The IABP has a positive impact on coronary artery blood flow perfusion in each of the experimental scenarios of LV heart failure, independent of the total cardiac output. We may be able to predict the role of IABP on coronary flow during VA-ECMO weaning by analyzing our results from an inverse timeline (Table 1 and Table 2).(3)The contribution of the IABP balloon to coronary artery perfusion at a constant flow rate (5 L/min) is less when the flow is primarily retrograde (supplied by the ECMO) than when the flow is entirely anterograde (absence of ECMO). One hypothesis is that the balloon may cause a resistance in the aorta that impedes the retrograde flow coming from the femoral artery ECMO cannula and that this occurs mainly during inflation and diastole (time of coronary artery perfusion). The present hypothesis could be investigated further using a model where the injecting cannula is placed in the right subclavian artery.

### Limitations

First, these results require further confirmation by additional in vitro mock-circuit experiments and clinical studies addressing this hypothesis. Second, an improvement in the mock-circuit setup to include physiological regulation of HR (mock baroreceptors), variable LV contractility (mock ejection fraction regulator), and the possibility to increase or decrease distal vasculature resistance in the circuit (mock preload/afterload) is currently in progress. These additions to the setup may help us to further deduce the expected benefits in aortic flow and coronary artery perfusion during the combination of ECMO with IABP in low cardiac output states. Third, we used the percentage of volume amplitude (100%) to give the best pressure curve resolution with the least resonance.

Finally, our results focused only on an IABP setting of 1:1, as we did not consider the 1:2 or 1:3 modes. The reason for this was the difficulty of synchronizing the IABP with the mock system, as the trigger used was a pressure threshold and not an ECG. Indeed, when 1:2 and 1:3 modes were used, back pressure waves and flow wave oscillations were erroneously considered as beats generated by the pulsatile pump, making the results harder to interpret.

## 5. Conclusions

Based on our study results, reduced coronary artery blood flow in experimental heart failure scenarios can be compensated for by ECMO support. This would seem to reinforce the rationale behind ECMO support in the treatment of ischemic and non-ischemic acute cardiogenic shock. Moreover, combining IABP with ECMO additionally improves coronary artery blood flow. This effect might positively impact heart failure recovery. Finally, the increase in coronary artery blood flow was more important in low heart rate conditions, most probably due to the increased time of inflation at heart rates of 60 bpm.

## Figures and Tables

**Figure 1 jcm-10-05333-f001:**
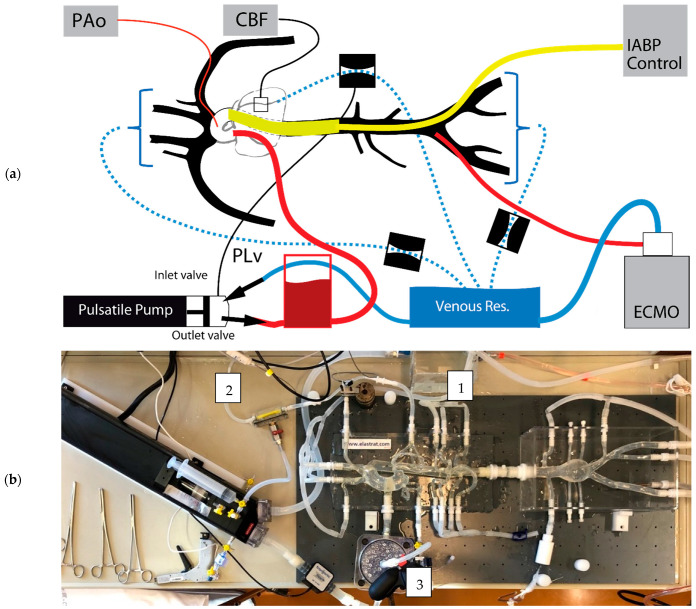
In vitro mock-circuit setup: schematic (**a**) and real (**b**). Measurements were made at the coronary level with a flowmeter (CBF) and in the central aorta with a pressure sensor (PAo). (1) Hemodynamic resistances were fixed with Hoffmann clamps. (2) Systolo-diastolic varying resistance synchronized to pump pulsatility, simulating the effect of left ventricular contraction (PLv). (3) Compliance reservoir. ECMO drained liquid from the venous tank and injected it into the femoral artery.

**Figure 2 jcm-10-05333-f002:**
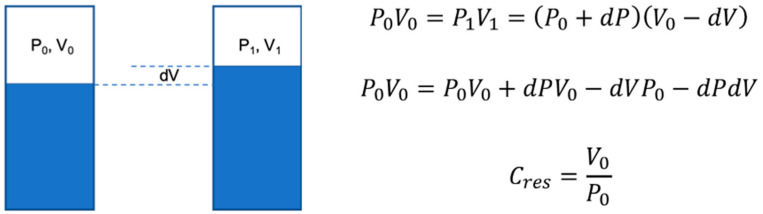
Compliance reservoir schematic in two different conditions to derive the equations. To obtain physiological pulse pressure (PP) at the aortic level, (V_0_) was adjusted. The added compliance (Cres) was 1.03 mL/mmHg at HR of 60/min and 0.31 mL/mmHg at HR of 100 bpm.

**Figure 3 jcm-10-05333-f003:**
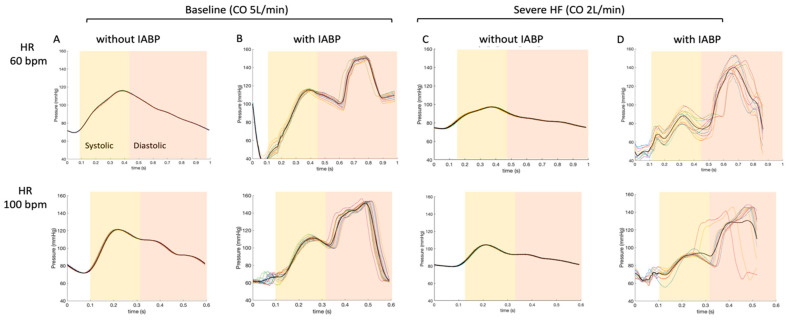
Average of pressure waveforms at the ascending aorta level for a heart rate of 60 bpm (top) and a heart rate of 100 bpm (bottom). Baseline conditions (CO of 5 L/min) with and without IABP enabled are presented in columns (**A**) and (**B**), respectively. Severe HF scenarios (CO of 2 L/min) with and without IABP enabled are presented in (**C**,**D**). The waveforms (thin lines) are recorded on 8–10 consecutive cardiac cycles and averaged (thick lines).

**Figure 4 jcm-10-05333-f004:**
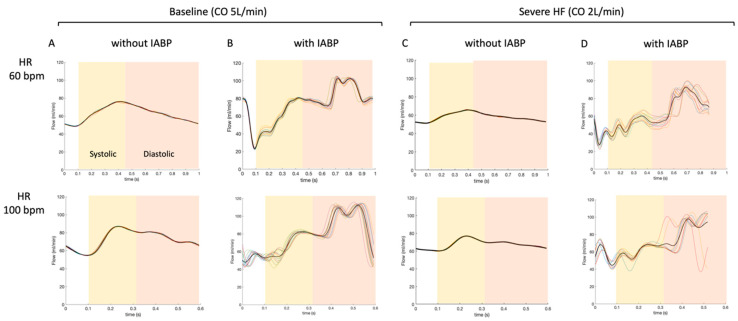
Average of flow waveforms at the coronary artery level for a heart rate of 60 bpm (top) and a heart rate of 100 bpm (bottom). Baseline conditions (CO of 5 L/min) with and without IABP enabled are represented in columns (**A**) and (**B**), respectively. Severe HF scenarios (CO of 2 L/min) with and without IABP are represented in (**C**,**D**). The waveforms (thin lines) are recorded on 8–10 consecutives cardiac cycles and averaged (thick lines). The varying resistance model of the coronary was not enabled in this case.

**Figure 5 jcm-10-05333-f005:**
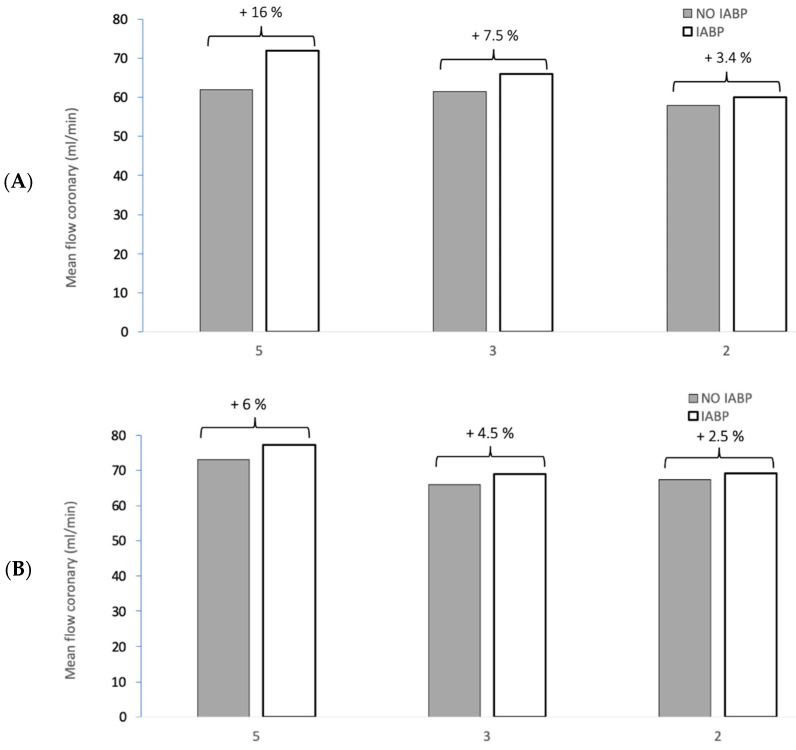
Coronary mean flow in different clinical scenarios: (**A**) (HR 60 bpm) and (**B**) (HR 100 bpm). The different clinical scenarios highlight the impact of IABP at different CO. The first situation corresponds to a baseline condition with a cardiac output of 5 L/min without ECMO support. Low CO syndrome by gradually decreasing CO to 2 L/min compensated to a total of 5 L/min with ECMO.

**Figure 6 jcm-10-05333-f006:**
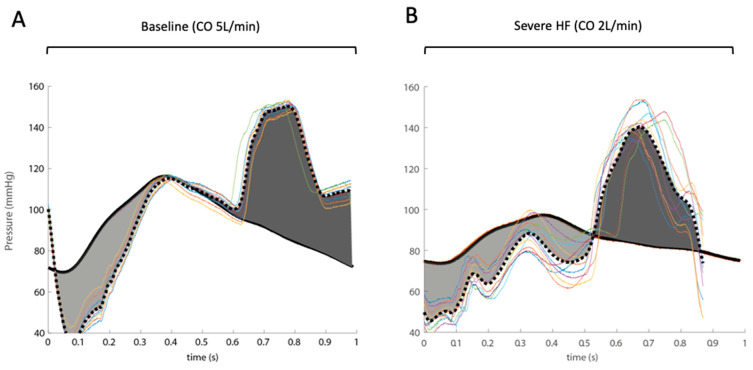
Pressure waveforms at the ascending aorta level at the baseline condition (CO 5 L/min) (**A**) and in the severe LV failure condition (CO 2 L/min) (**B**) are presented for a heart rate of 60 bpm (continuous lines). The effect of enabling the IABP on the pressure waveforms is superposed (dashed line) and is highlighted by the area under the curves (AUC) emphasizing the pressure decrease during systolic and the increase during diastolic compared to the pressure waveform without enabling the IABP.

**Table 1 jcm-10-05333-t001:** A summary of the mean flow and pressure measurements for the different scenarios of progressive heart failure is presented for an HR of 60 bpm. For each scenario, data are reported whether IABP is enabled (100%) or not. The mean flow increase in the coronary is relative to the reference value without the IABP (%). The generated ECMO flow is represented. The varying resistance model of the coronary was not enabled in this case.

Pump Output (L/min)	5		4		3		2	
ECMO (L/min)	-		1		2		3	
IABP function	-	100%	-	100%	-	100%	-	100%
Mean coronary flow (mL/min)	62	72	65.6	70.4	61.5	66.1	58	60
Mean coronary flow increase (%)		16		7.3		7.5		3.4
Mean pressure aortic arch (mmHg)	93	101	92	99	87	94	85	89
Mean ascending aorta flow (mL/min)	4859	4832	3982	3984	3039	3078	2045	2036

**Table 2 jcm-10-05333-t002:** A summary of the mean flow and pressure measurements for the different scenarios of progressive heart failure is presented for an HR of 100 bpm. For each scenario, data are reported whether IABP is enabled (100%) or not. The mean flow increase in the coronary is relative to the reference value without the IABP (%). The generated ECMO flow is represented. The varying resistance model of the coronary was not enabled in this case.

Pump Output (L/min)	5		4		3		2	
ECMO (L/min)	-		1		2		3	
IABP function	-	100%	-	100%	-	100%	-	100%
Mean coronary flow (mL/min)	73	77.4	68	71.4	66	69	67.4	69.1
Mean coronary flow increase (%)		6		5		4.5		2.5
Mean pressure aortic arch (mmHg)	98	102	91	95	89		90	95
Mean ascending aorta flow (mL/min)	5046	5017	4012	4007	3076	3038	1997	2039

## Data Availability

Data supporting the findings of this study are available within the article.

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
