# Peer review of "To Balloon or Not to Balloon? The Effects of an Intra-Aortic Balloon-Pump on Coronary Artery Flow during Extracorporeal Circulation Simulating Normal and Low Cardiac Output Syndromes"

_jcm, 2021, doi:10.3390/jcm10225333_

Round 1

Reviewer 1 Report

In this study, entitled “To Balloon or not to Balloon! The Effects of the Intra-Aortic Balloon-Pump on Coronary Artery Flow during Extra Corporeal Circulation Simulating Normal and Low Cardiac Output Syndromes”, the authors present a study evaluating the effect of IABP on coronary artery flow during ECMO in a HF setup using an in-vitro circuit, reporting that IABP improved coronary blood flow, particularly at a lower heart rate. The study is interesting, the topic is of interest; the manuscript is well structured and well written.

However, there are some issues:

- In the Results, I would describe more extensively the changes in coronary blood flow with IABP (absolute numbers and percentages) in the different scenarios.

- The Discussion could be improved. I would minimize the description of IABP functioning (first part of the Discussion) and I would prioritize the contextualization of the study findings in available evidence. The present study was performed in an in-vitro setting; hence, a translational outlook is needed, and the findings need to be commented with a concomitant focus on the clinical setting. In this sense, recent IABP reviews could be mentioned and discussed (doi: 10.1161/CIRCHEARTFAILURE.121.008527; doi: 10.1007/s11897-020-00480-0). Similarly, clinical studies reporting a better response to IABP in patients without tachycardia could be reported and discussed (PMID: 29335384), since this is in line with a relevant study finding.

Author Response

Dear Reviewer,

We are very grateful for your thorough reviews.

- In the Results, I would describe more extensively the changes in coronary blood flow with IABP (absolute numbers and percentages) in the different scenarios.

We have taken your relevant comment into account and rewritten the paragraph on coronary flow results in more detail.

Unfortunately, Fig. 5 b) representing the histograms of coronary mean flow for the clinical scenarios of heart rate 100 bpm was not visible in the manuscript. The figure has been updated.

- The Discussion could be improved. I would minimize the description of IABP functioning (first part of the Discussion) and I would prioritize the contextualization of the study findings in available evidence.

The present study was performed in an in-vitro setting; hence, a translational outlook is needed, and the findings need to be commented with a concomitant focus on the clinical setting. In this sense, recent IABP reviews could be mentioned and discussed (doi: 10.1161/CIRCHEARTFAILURE.121.008527; doi: 10.1007/s11897-020-00480-0).

Similarly, clinical studies reporting a better response to IABP in patients without tachycardia could be reported and discussed (PMID: 29335384), since this is in line with a relevant study finding.
Thank you for your relevant comments. As suggested we have reduced the description of the IABP balloon in the discussion. The references on the clinical results of the use of the balloon that you mentioned are very interesting, we have added them in the introduction, in the method section 2.4 and in the discussion with the corresponding text.

Reviewer 2 Report

The authors have submitted an in vitro-study focusing on the efficacy of ECMO in increasing the coronary blood flow in patients supported by VA-ECMO. A sophisticated model was constructed, simulating the aortic vessel flow and resistance, the coronary flow and resistance, the IABP and the ECMO blood flow.

Several flow conditions were considered during hemodynamic measurements in order to simulate normal cardiac output, moderate and severe heart failure.

I have some comments that should be useful to further improve this already interesting research.

1) IABP was set-up in semi-automatic mode 1:1. Why the authors did not consider coronary blood flow at 1:2 or 1:3 set-up? If they had some difficulties in such analysis, this should be state in discussion and/or limitation.

2) Was the IABP set-up with 100% volume? And, as above, why the authors did not provide measurements at different volume percentage (i.e. 70%, 50%)

3) An interesting issue could be to investigate the role of IABP on coronary blood flow during VA-ECMO weaning. Please make some comments in discussion.

4) Figures 3 and 4 should be improved adding a flow diagram related to severe HF without IABP.

Overall this is a timely and relevant research in-vitro study that may give important answers to the debated use of IABP during VA-ECMO

Author Response

Dear Reviewer,
We are very grateful for your thorough reviews.
1) Thank you for your valuable comments.
We had considered the 1:2 mode in preliminary experiments when we were performing our experiments. However, when 1:2 mode was used it was difficult to synchronize the IABP to the mock circuit-setup as the trigger used was a pressure threshold and not ECG. Indeed, back pressure waves and flow wave oscillations in our experimental setup were considered by the model as a beat generated by pulsatile pump. In other words, we were using a trigger based on pressure measurement, this made the adjustment of the balloon control more delicate and more variable because it depended on the oscillations/fluctuations of this pressure measured in the ascending aorta. The results were even more difficult to interpret.
We mention this limitation in the discussion.
In a later improved version of our experimental setup, we plan to "trigger" the IABP control based on an "ECG" signal as frequently encountered in clinical settings. This generic ECG signal would be generated and synchronized on the pulsatile pump control signal, which would allow us to get rid of these difficulties and allow us to conduct new experiments in order to study the cases of figures (mode 1:2 and 1:3) that you mentioned.
2) Thank you for your relevant comments on another IABP parameter, we used the percentage of volume amplitude giving the best pressure curve resolutions, with the least resonance.
We mention this sentence in the limitation as well and added de volume value in the method section 2.4.
3) We are not sure to have understand your question, as it seems to us, that the response to the present question is highlighted in our experiment if we inverse the timeline of the experiment. We have added a sentence highlighting this fact in the discussion p.9.
4) Thank you for this relevant comment that improves the understanding. We have added the pressure and flow curves corresponding to the severe HF scenario without IABP to figures 3 and 4 respectively.